# Mycotoxins in Feed: Hazards, Toxicology, and Plant Extract-Based Remedies

**DOI:** 10.3390/metabo15040219

**Published:** 2025-03-24

**Authors:** Xiangnan Zhang, Jiashun Chen, Xiaokang Ma, Xiongzhuo Tang, Bie Tan, Peng Liao, Kang Yao, Qian Jiang

**Affiliations:** 1Animal Nutritional Genome and Germplasm Innovation Research Center, College of Animal Science and Technology, Hunan Agricultural University, Changsha 410128, China; xiangnan_zhang2022@163.com (X.Z.); jschen@hunau.edu.cn (J.C.); maxiaokang@hunau.edu.cn (X.M.); txiongzhuo@163.com (X.T.); bietan@hunau.edu.cn (B.T.); 2Yuelushan Laboratory, Changsha 410128, China; 3Institute of Subtropical Agriculture, Chinese Academy of Sciences, Changsha 410125, China; liaopeng@isa.ac.cn

**Keywords:** mycotoxins, fungal secondary metabolites, plant extracts, livestock production

## Abstract

**Background:** Mycotoxins, which are secondary metabolites produced by fungi, are prevalent in animal feed and pose a serious risk to the healthy growth of livestock and poultry. **Methods:** This review aims to conclude current knowledge on the detrimental effects of mycotoxins on animal health and to demonstrate the potential of plant extracts as a means to counteract mycotoxin toxicity in feed. A systematic review of the literature was conducted to identify studies on the impact of mycotoxins on livestock and poultry health, as well as research into the use of plant extracts as feed additives to mitigate mycotoxin effects. Studies were selected based on their relevance to the topic, and data were extracted regarding the mechanisms of action and the efficacy of plant extracts. **Results:** Excessive mycotoxins in feed can lead to reduced appetite, impaired digestion, and general health issues in animals, resulting in decreased food intake, slowed weight gain, and instances of acute poisoning. Plant extracts with antioxidant, anti-inflammatory, and anti-mutagenic properties have shown the potential to improve production efficiency and reduce the toxic effects of mycotoxins. **Conclusion:** This comprehensive review not only consolidates the well-documented adverse effects of mycotoxins on animal health but also introduces a novel perspective by highlighting the potential of plant extracts as a promising and natural solution to counteract mycotoxin toxicity.

## 1. Types of Mycotoxins and Current Status of Feed Contamination

Mycotoxins, which are toxic compounds produced by specific fungi, pose a serious threat of contamination to a wide range of food sources. In agricultural products, molds are capable of producing more than 300 types of mycotoxins, many of which are considered major threats to human and animal health, with toxic and carcinogenic effects on the immune system, liver, kidneys, genes, and reproductive system [1]. The presence of these toxins in feed involves multiple complex steps, including fungal growth, toxin production, contamination pathways, absorption, distribution, and ultimate toxic effects. Ranging from hepatotoxicity to immunotoxicity, these toxins severely impact animal health, and due to their thermal stability, they can survive during feed processing, posing a persistent threat to animal productivity (Table 1). Therefore, a deep understanding of the mechanism of action of mycotoxins is crucial for developing effective prevention and control strategies to mitigate their polluting impact. Against this backdrop, measures for the prevention and control of mycotoxins are vital, and post-contamination detoxification technologies are equally critical. Recently, plant extracts containing flavonoids, terpenoids, and phenolics have become the focus of research on mycotoxin control and detoxification, with the potential of these natural compounds lying in their ability to interfere with the absorption and bioavailability of toxins, potentially reducing the toxicity of mycotoxins [2].

### 1.1. Types and Chemical Stability of Mycotoxins

Most mycotoxins are produced by toxigenic molds, such as *Aspergillus*, *Penicillium*, *Fusarium*, *Claviceps*, and *Grapevine*, and certain species of endophytic fungi. At present, more than 300 kinds of mycotoxins are known, among which aflatoxins (AFs), deoxynivalenol (DON), zearalenone (ZEA), fumonisins (FBs), and ochratoxin-A (OTA) are serious hazards to animal husbandry. They are widely distributed in cereals, such as maize and rice, and nuts, such as pistachios. These mycotoxins have stable physical and chemical properties and are resistant to high temperatures during conventional feed processing (including heating). For example, aflatoxin B1 (AFB1) is stable at most feed processing temperatures and has a boiling point of 268 °C [3]. DON remains chemically stable through processing, storage, and even cooking, which makes it difficult to eliminate [4].

**Table 1 metabolites-15-00219-t001:** Five common mycotoxins in feed and their mechanisms of toxicology.

Mycotoxin Types	Sources	Toxicity Characteristics	Mechanisms	Reference
AFs	*Aspergillus flavus* and *Aspergillus parasiticus*	Highly toxic, hepatotoxic and carcinogenic	Inhibits protein synthesis, causes cell apoptosis, and increases the risk of hepatic carcinogenesis	[5]
DON	*Fusarium graminearum*	Food refusal, vomiting, growth retardation, and immunosuppression	Activates the intracellular stress response, leading to apoptosis	[6]
ZEA	*Fusarium graminearum* and *Fusarium rhubarb*	Simulating estrogen, pseudopregnancy, vaginitis, and infertility	Activates estrogen receptors and affects the development of reproductive organs	[7]
FBs	*Fusarium* sp.	Neurotoxicity, liver and kidney injuries, immunosuppression, and reproductive disorders	Inhibits sphingolipid biosynthesis and disrupts cell membrane structure	[8]
OTA	*Aspergillus* spp. and *Penicillium* spp.	Kidney injury, immunosuppression, and growth retardation	Inhibits protein synthesis and causes cell death	[9]

### 1.2. Widespread Occurrence of Mycotoxin Contamination

Mycotoxin contamination is a major challenge to feed safety globally, affecting 60–80% of agriculture products. The prevalence, co-occurrence patterns, and regulatory exceedances are listed in Table 2. A quarter of the world’s food and feed crops are susceptible to mycotoxins, according to the FAO (The Food and Agriculture Organization). The European survey in 2021 showed that 98.5% of nearly 1200 feed samples contained at least one mycotoxin, 86.1% contained two or more mycotoxins, and 77% of cereals such as wheat and barley were detected with novel mycotoxins [10]. AFs are widely present in the feed of Indian dairy cattle, with AFs detected in 59% of concentrated feeds, 44% exceeding the US limit of 20 μg/kg, and 58% exceeding the EU limit of 5 μg/kg [11]. The study conducted by Zhao et al. in China from 2018 to 2020 found that the average content of AFB1 in feed and whole feed was 1.2–27.4 μg/kg, the incidence rate was 81.9–100%, and 0.9% of the raw materials were contaminated with AFB1 exceeding the Chinese safety standard concentration [12]. Wei Hao et al. ’s study, covering 9392 feed samples in China from 2017 to 2021, found that trichothecene B and FB contamination were the most common in new season maize, with detection rates as high as 84.04% and 87.16%, respectively [13]. Zhang Yong and his colleagues analyzed 1025 feed samples and found that the detection rates of AFB1, ZEN, and DON reached 95.99%, 98.54%, and 100%, respectively. Particularly striking, 89.85% of the samples contained these three mycotoxins, revealing a widespread problem of multiple contamination in 2021 [14]. Cargill Animal Nutrition analyzed over 350,000 results from 145,000 raw material samples collected from more than 150 feed mills, farms, and raw material storage sites in 2023, which indicated that the top three in terms of prevalence and risk level were DON, FBs, and ZEA [15]. Based on a globally extensive survey, it has been revealed that climate is a crucial factor in determining the areas contaminated by mycotoxins. In several regions of South Asia, Sub-Saharan Africa, and Southeast Asia, rainfall or significant temperature variations during the critical period of grain development are the decisive factors for the concentration of mycotoxin contamination. A large number of samples (64%) were co-contaminated by ≥two types of mycotoxins [16]. A study examined data from 717 wheat fields across Norway, Sweden, Finland, and the Netherlands. The observational data included details on flowering dates, the length of time between flowering and harvest, the resistance of wheat to *Fusarium* infection, and the concentration of deoxynivalenol (DON) in relation to various climate variables critical to wheat cultivation, such as relative humidity, temperature, and rainfall. The findings highlighted that the pattern of DON contamination in wheat in Northwestern Europe is influenced by climatic changes [17]. Taken together, mycotoxin contamination poses a global challenge that requires continuous surveillance.

**Table 2 metabolites-15-00219-t002:** Mycotoxin contamination in feed: prevalence, co-occurrence patterns, and regulatory exceedances.

Mycotoxin Type	Country/Region	Study Period/Year	Sample Size/Scope	Contamination Levels	Detection Rate (%)	Exceedance of Limits
Multiple mycotoxins	Europe	2021	~1200 feed samples	98.5% with ≥1 mycotoxin; 86.1% with ≥2 mycotoxins; 77% cereals with novel mycotoxins	98.5 (any)	-
AFs	India	-	Dairy cattle feed	AF detected in 59% of concentrated feeds	59	44% exceeded US limit (20 μg/kg); 58% exceeded EU limit (5 μg/kg)
AFB1	China	2018–2020	Various feeds	Average content: 1.2–27.4 μg/kg; incidence: 81.9–100%	81.9–100	0.9% of materials exceeded the Chinese safety standard
Trichothecene B and FBs	China	2017–2021	9392 feed samples	Most common in new season maize	84.04 (trichothecene B); 87.16 (FBs)	-
AFB1, ZEN, and DON	China	2021	1025 feed samples	Co-contamination prevalence	95.99 (AFB1); 98.54 (ZEN); 100 (DON)	89.85% of samples contained all three mycotoxins
DON, FBs, and ZEN	Global (Cargill data)	2023	>145,000 samples from 150+ sites	Top 3 prevalent mycotoxins by risk level	-	Global contamination patterns observed

### 1.3. Effects and Toxicology of Five Common Mycotoxin Contaminants on Livestock and Poultry

Ingestion of feed contaminated with these mycotoxins commonly induces poisoning in animals. Low concentrations impair animal growth performance and immune function, resulting in liver and kidney dysfunctions, intestinal syndrome, digestive tract inflammation, and reproductive dysfunction, while high concentrations cause acute death in animals [18,19]. In addition, the residues of mycotoxins in meat, eggs, milk, and other products have become a major hazard to the safety of animal-derived food [20]. The effective management and mitigation of mycotoxin contamination in food and feed play a crucial role in improving animal performance and ensuring the safety of the human food chain. The pathological characteristics of different animals exposed to these five common mycotoxins in diets are listed in Table 3.

AFs, as a high-risk and prevalent mycotoxin, pose significant threats to livestock health and productivity [21]. Aflatoxin mainly affects the livers of animals, and chronic exposure can cause fatty liver degeneration and liver tissue necrosis. The characteristics of poisoning include depression, loss of appetite, weight loss, stagnating movement, and decreased immunity. Liver damage was particularly significant, which was manifested as liver enlargement, fragile texture, and brown color and, in severe cases, accompanied by liver fibrosis and pathological structural changes [22,23,24,25]. Although pigs are slightly less sensitive to aflatoxin than birds, their liver damage is more serious, and the ingestion of contaminated feed by sows can be transmitted to piglets through milk, leading to growth retardation or death [26,27]. In addition, the ingestion of aflatoxin-contaminated diets by cows can destroy their intestinal barrier function, affect their immune response, and weaken their defense against pathogens while impairing their cellular and humoral immune mechanisms and reducing their resistance to viral infections and parasites [28].

ZEA, as a kind of estrogen substance, mainly affects the function of the reproductive system and can cause estrogen hyperemia in animals. The intake of ZEA during pregnancy can cause abortion, stillbirths, and malformation and has negative effects on the central nervous system, such as nausea, chills, headache, confusion, and coordination disorders [7]. Gilts are particularly sensitive to ZEA, and the typical characteristic of poisoning is vulvovaginitis with abnormal mammary gland development, while boars show decreased libido, testicular atrophy, and decreased semen quality [29]. Compared with swine, poultry has a higher tolerance to ZEA due to its efficient hepatic and intestinal circulation, rapid excretion capacity, and specific physiological characteristics, such as microbial transformation, differential enzyme activities, and low estrogen receptor binding capacity [30]. Ruminants are less susceptible to ZEA, which may be attributed to their rumen microorganisms degrading ZEA.

**Table 3 metabolites-15-00219-t003:** Pathological characteristics of different animals exposed to five common mycotoxins in diets.

Mycotoxin Types	Effects on Monogastric Animal	Effects on Poultry	Effects on Ruminants	Effects on Aquatic Livestock
AFs	Growth retardation, decreased feed utilization, depression, anorexia, acute liver disease, and immunosuppression	Subcutaneous hemorrhage, smaller eggs, reduced yolk weight, and reduced fertilization and hatching rate	Reduced resistance of cows to viruses and parasites	Liver necrosis, decreased feed intake, and weight loss were observed
ZEA	Pseudoestrus, vaginitis, abortion, stillbirth in gilts, testicular atrophy in boars, and decreased semen quality in gilts	The ovaries were atrophic, the egg production rate was decreased, and the fertilization rate of breeding eggs was decreased	The rumen degrades 90% of ZEA and generates the more toxic zearalenol	Decreased fecundity, ovulation disorders, and infertility
DON	Decreased feed intake, intestinal damage, vomiting, and food refusal	Invasion of the digestive tract, decreased feed intake, food refusal, and reduced egg production rate	Under stress, the risk of poisoning is increased	Destroying the integrity of the gill structure
FBs	Growth arrest, liver tissue damage, reproductive impairment, and immunosuppression	Reduced egg production, poor feather growth, oral ulcers, and neurological disorders	Weight loss, immunosuppression, and liver toxicity	Inhibit growth and cause pathological damage
OTA	Weight loss, growth retardation, and liver and kidney lesions	Incomplete eggshell calcification, high rate of egg breaking, and subcutaneous hemorrhage	Anorexia, diarrhea, difficulty gaining weight, and decreased milk production	Nervous system and respiratory toxicities

DON has been identified as one of the most dangerous natural food contaminants by the FAO and WHO. After the ingestion of feed contaminated with DON, livestock will show signs of anorexia, vomiting, diarrhea, fever, dyskinesia, and delayed response. In severe cases, it can lead to blood system damage and even death [31]. Pigs are the most sensitive to DON, showing obvious loss of appetite, food refusal, vomiting, and intestinal damage. Every 1 mg/kg of DON in the diet can reduce the weight gain of pigs by 8% [32]. Moreover, 2 mg/kg of DON in the diet significantly decreased the digestibility of essential amino acids, impaired the absorption of nutrients, and reduced the utilization rate of energy and nutrition [33]. Birds have a high tolerance to DON; however, high doses of DON also reduce their feed intake and growth performance [34,35]. Ruminants show a high tolerance to DON; likewise, beef cattle and sheep can tolerate up to 21 mg/kg of DON intake [36,37,38].

FBs commonly cause pulmonary edema; damage the liver, kidneys, and intestine; affect the reproductive health of boars; reduce sperm quality; and interfere with the ovarian function of sows [39,40,41,42]. For example, 2 μg/kg injection of FB1 into the air cell of chicken eggs can significantly reduce the number of chicken embryos and cause bleeding [43]. The air chamber experiment of chicken embryos incubated for 72 h after injection of different doses of FB1 showed that the mortality rate of chicken embryos was up to 100% [44].

The action mechanism of OTA on ruminants and monogastric animals is different: the former is degraded by microorganisms in the rumen, while the latter is absorbed directly in the intestine [45]. OTA mainly attacks the kidney, showing nephrotoxicity in all monogastric animals, resulting in kidney atrophy, enlargement, and pallor [46]. Experiments have shown that piglets fed rice cultures contaminated with OTA (0.2 to 0.6 mg/kg body weight per day) or pure toxin (2.0 or 1.0 mg/kg body weight per day) suddenly develop depression and reduced feed intake, resulting in weight loss, followed by diarrhea, anorexia, and dehydration [47]. After consuming OTA-contaminated feed, 65% of OTA was absorbed in the GI tract, and the remaining OTA was deposited in the kidney, resulting in weight loss, low feed efficiency, and weight gain [48]. Physiological symptoms of OTA toxicity in poultry include fatigue, reduced feed consumption, developmental delay, poor oviposition, reduced plumage, and extreme mortality at high dietary concentrations, with weight gain reduced in a dose-dependent manner when infected at rates of 0.5 to 29.4 mg/kg for 7 to 60 days in experimental studies. Reduced feed intake was observed between 21 and 60 days of 0.5–4 mg/kg OTA feeding, whereas egg production, hatchability, shell thickness, and egg production were severely affected between 28 and 84 days of 0.5–4 mg/kg OTA feeding in the diet [48,49]. In cattle, microbes in the rumen degrade low doses of OTA to nontoxic forms. Higher doses of OTA ingestion by cattle cause anorexia, diarrhea, difficulty gaining weight, and rapid loss of milk production [50].

In conclusion, different types of mycotoxins have different hazards to different livestock and poultry, and prevention and control can be focused according to the category of livestock and poultry.

## 2. Mechanisms and Targets of Feed Mycotoxins That Harm the Health and Growth of Livestock and Poultry

### 2.1. AFs

AFs are the most frequent mycotoxins in human food and animal feed. Their fat-soluble properties promote AFs to invade organisms through the skin, respiratory tract, and digestive tract, especially the digestive tract is the main pathway for the absorption of feed mycotoxins [51,52]. According to a previous study, about half of AFB1 from feed is absorbed in the duodenum and then transported to the target tissues through the blood circulation and lymphatic system [53].

The liver plays a central role in the biotransformation of AFs, in which AFB1 undergoes hydroxylation, epoxidation, hydration, and α-dimethylation catalyzed by the reductase system in the liver microsome and cytoplasm to be converted to less toxic derivatives. This conversion process includes reversible and irreversible reaction types [54,55,56].

Aflatoxin B1 is converted to a more active B1–8, 9-epoxide catalyzed by the hepatic microsomal P450 enzyme system. This epoxide can form stable adducts to DNA and proteins, for example, by covalent bonding at position C8 to DNA guanine N7 and binding to lysine residues, interfering with genetic information transmission and protein synthesis. It can induce genetic mutations, malformations, and carcinogenic effects [57,58].

By promoting the high-frequency mutation of the p53 gene, aflatoxin B1 not only eliminates its tumor suppressor function, but also activates the anti-apoptotic pathway, drives cell proliferation disorders, and then causes tumor formation, showing the typical characteristics of oncogenes. These mutant p53 proteins also tend to build stable complexes with oncogenic proteins, prolong their intracellular stability and over-express, and accelerate the process of cell carcinogenesis [59].

As a powerful immunosuppressant, aflatoxin has multiple effects, not only hindering the normal development of immune organs but also weakening host defense by inhibiting humoral and cellular immune mechanisms and increasing the sensitivity to bacterial, viral, and parasitic infections [60,61]. Aflatoxin also inhibits protein synthesis through DNA/RNA binding, resulting in thymic degeneration, the depletion of lymphocytes, and the impairment of liver and macrophage functions, accompanied by the suppression of complement C4 and T-lymphokine production [62,63]. At the level of humoral immunity, aflatoxin interferes with the activity of RNA polymerase and the synthesis of immunoglobulin, thus showing immunotoxicity [52]. In terms of cellular immunity, aflatoxin B1 further weakened the cellular immune response by downregulating the mRNA expression of IL-2 and IFN-γ in hepatocytes [64].

### 2.2. ZEA

The absorption and biotransformation sites of ZEA are the gastrointestinal tract and liver. Its metabolic pathway can be summarized as a two-stage process: The first stage involves the hydroxylation reaction. Under the catalysis of steroid hydrogenase and 3α/3β-hydroxysteroid deoxy enzyme, ZEA undergoes ketone group reduction at the C-6 position to produce α- and β-ZEA isomers and then undergoes double-bond reduction at the C11-C12 position to form α- and β- ZEA. In the second stage, uridine diphosphate glucuronosyltransferase (UDP-glucuronosyltransferase) catalyzes glucuronic acid conjugation of these compounds, resulting in the resultant conjugates being cleared from the body through different excretion pathways [65,66,67].

Studies have shown that ZEA and its derivatives ZOL and ZEL can specifically and competitively bind to the ERα and ERβ isoforms of the ER (estrogen receptor) in the cytoplasm, mimicking the mechanism of 17-β-estradiol. The structure of the phenolic ring in ZEA and its derivatives ZOL and ZEL show consistency with the binding domain of the A ring of estradiol in the active site. Notably, as a full agonist of ERα and a partial agonist of ERβ, the modulatory effects of ZEA on estrogenic activity are mainly mediated through ERα. After binding, the ER-ZEA complex is transported into the nucleus and interacts with estrogen response elements (EREs) to regulate target gene transcription and protein synthesis, interfere with cell proliferation and growth, and display typical estrogen-like activities [68,69,70].

The mechanism by which ZEA promotes ROS production may involve structural changes in mitochondria. As the main source of ROS, ZEA can reduce mitochondrial membrane potential, mitochondrial membrane damage, permeability transition, and cytochrome c oxidase inactivation in a dose-dependent manner. The inhibition of immunoglobulin secretion may result from the direct inhibition of B cells, reducing immunoglobulin production, given that plasma cells are responsible for immunoglobulin secretion after differentiation from B cells upon antigen stimulation [71,72].

### 2.3. DON

DON is absorbed primarily in the duodenum, which, thanks to its small-molecule properties, allows for passive transport into the osmotic bloodstream. Generally, DON interferes with the function of 5-hydroxytryptamine and catecholamine receptors in the small intestine and interferes with the normal peristaltic rhythm, resulting in a series of digestive disorders, such as nausea, vomiting, loss of appetite, and diarrhea [73].

Studies have shown that DON can reduce the transepithelial electrical resistance (TEER) of intestinal epithelial cell monolayer in a dose-dependent manner. In addition, DON downregulated the expression of tight junction proteins (ZO-1, E-cadherin, Occludin, and Claudin) in the intestinal epithelium both in vitro and in vivo, which may be regulated by the activation of the MAPK pathway [74,75]. DON has also been found to significantly increase the expressions of TNF-α, IL-1β, IFN-γ, IL-6, and other pro-inflammatory factors in the jejunums and ilea of piglets, affect cytokine and immune function, and damage the intestinal mucosal barrier [76]. Additionally, DON affects mucosal SIgA secretion and further damages the host defense against pathogens, indicating the adverse effects of DON on intestinal mucosal immunity [77].

### 2.4. FBs

The gastrointestinal absorption of FBs in pigs is limited, and the absorption rate of fumonisin-B1 (FB1) is only about 4%. The accumulation of FB1 was mainly distributed in the liver, kidneys, and spleen in pigs, and its distribution pattern was significantly affected by the dose and duration of exposure [78]. Specifically, during low-dose, long-term exposure, FB1 accumulated the most in the lungs, followed by the liver and kidneys. On the other hand, when a high dose was taken for a short period, the accumulation of the toxin in the liver was the first, followed by the kidneys, heart, and lungs, showing the regulation of the dose and time on the distribution of the toxin [79].

FB1, which is structurally similar to sphingosine, can inhibit the activity of N-acyltransferase, thereby interfering with sphingomyelin synthesis [80]. Sphingolipids, as a key component of the cell membrane, are involved in the regulation of cell life activities. FB competes with sphingosine N-2 acyltransferase due to its structural similarity, hindering sphingolipid metabolism and causing cell damage [81]. In addition, by interfering with sphingolipid-mediated cell signaling, FB1 can affect the activities of protein kinase C, phosphorylase D, and AP-1 transcription factors and interfere with cell adhesion, differentiation, proliferation, and even carcinogenesis [82,83].

Several studies have suggested that FB1 may mediate toxicity and carcinogenicity through oxidative stress. For example, when HepG2 cells were treated with 50 μM FB1 for 0, 12, and 24 h, the ROS content in treated cells was significantly higher than that in blank cells [42]. When 2.5 mg/kg FB 1 was exposed to the colon of mice, antioxidant enzymes such as SOD-1, SOD-2, GR, and GADPH in the FB 1-treated group were significantly lower than those in the control group. The contents of CYTP450, TRX, HSP70, and HSP90 were significantly higher than those in the control group [84].

### 2.5. OTA

OTA is mainly absorbed in the small intestine segment and undergoes a biotransformation process in the liver and kidneys of pigs [85]. After entering the blood, the OTA binds serum proteins in a bivalent form and maintains its long-term toxic effect through interaction with biomacromolecules and reabsorption in the kidneys and hepatic–intestinal circulation [86].

OTA inhibits protein synthesis by competitively binding to tRNA synthesis sites and affects phenylalanine access, thereby inhibiting t-RNA function [87]. Its promotion of lipid peroxidation may be related to the activation of NADPH and the enhancement of ascorbate-dependent lipid peroxidation, leading to the accumulation of malondialdehyde [88]. In the main target organ, the kidneys, OTA inhibits renal enzyme activity through the phenylalanine group, affecting renal function. OTA can also cause liver peroxidation, leading to hepatocyte apoptosis and dysfunction [89]. Its cytotoxic mechanism may involve competitive inhibition of mitochondrial carrier proteins, blocking mitochondrial phosphate transport and the respiratory chain and affecting mitochondrial function. In addition, OTA inhibits protein synthesis, reduces immunoglobulin secretion, reduces the activity of immune factors, and inhibits the proliferation and apoptosis of immune cells [90].

## 3. Study on the Application of Plant Extracts in Mycotoxin Poisoning

### 3.1. The Development Potential of Plant Extracts

Since ancient times, plants have played a central role in the prevention and treatment of diseases, a tradition that continues to this day. Despite the remarkable advances in modern medicine, many people in developing regions still rely on traditional remedies to cope with health problems, both for cultural heritage, economic, and efficacy reasons [91]. According to the World Health Organization, about 80% of the world’s population relies on traditional medicine for their basic medical needs, demonstrating the universal value of plant medicine [92]. Recent studies have witnessed a resurgence of interest in medicinal products derived from plants in both developed and developing societies, highlighting the vast resources and geographical diversity of plants for medicinal use, as well as the translation of traditional wisdom into modern medicines derived from plants [93].

### 3.2. Source and Composition of Plant Extracts

Plant extracts are the products of physical extraction or biological fermentation of the whole or parts of one or more natural plants as raw materials. They are usually derived from the roots, stems, leaves, flowers, fruits, or seeds of plants and contain a variety of phytochemicals. According to the main active ingredients, they can be divided into alkaloids, glycosides, flavonoids, organic acids, polysaccharides, polyphenols, and volatile oils [94]. Saponins are widely distributed in higher plant kingdoms and play important roles in Chinese herbal medicine, such as ginseng, Astragalus membranaceus, Panax notoginseng, and jujube seed. They show a variety of pharmacological activities, including anti-cancer, anti-tumor, antipyretic, detoxification, etc., which can play a beneficial role in the immunity, antioxidant digestive, and metabolic functions of livestock and poultry, thereby improving the performance of livestock and poultry [95]. Flavonoids, which are widely distributed in plants, play key roles in plant growth and development and have antibacterial and disease-prevention functions [96]. As widely distributed non-amino acid compounds, organic acids are widely present in plants with acidic characteristics, especially in fruits. Su Hong et al. used the Box–Behnkenken design to optimize the compound biological preservative of citric acid, thymol, and sodium alginate and provided a natural and efficient microbial inhibition scheme [97]. Polysaccharides are mainly distributed in plants, microorganisms, and marine organisms, especially in plant tissues. ZHAI et al. showed that pomegranate peel polysaccharides can significantly improve the activity of antioxidant enzymes in mice with CCI4-induced liver damage, suggesting their protective effect [98]. Hu Aijun’s team extracted polysaccharides from chickpeas with the assistance of ultrasound, which proved that both non-starchy and acidic polysaccharides had free radical scavenging ability, emphasizing the potential of polysaccharides in biological protection and treatment [99]. Polyphenols are a class of secondary metabolites that are widely present in plants, and most representatives of these compounds are characterized by interactions with reactive oxygen species [100]. In summary, the active components of the plant extracts were proven to have significant antioxidative and tissue-protective effects, showing potential to ameliorate mycotoxin-induced damage. Polyphenolic compounds, flavonoids, and other bioactive components in plant extracts can prevent the absorption of mycotoxins or reduce their biological activity by binding to mycotoxin molecules. Phenolics from beetroot may interfere with the absorption and bioavailability of mycotoxins in the gastrointestinal tract, potentially reducing their toxicity [101]. Pumpkin extract plays a significant role in mitigating the toxic effects of aflatoxin B1 and Ochratoxin A on neuronal differentiation. Specifically, the addition of pumpkin extract in contaminated breads significantly reduced the bio-accessibility of aflatoxin B1 and Ochratoxin A [102]. Moreover, some extracts can enhance the detoxification capacity in animals by regulating the activities of detoxifying enzymes. This study indicates that the phenolic compounds in algal extracts can aid in the detoxification of rats damaged by AFB1 [103].

### 3.3. Application of Plant Extracts

Plant extracts are rich in bioactive components with anti-mutagenic, antioxidant, and anti-cancer properties, which may effectively reduce the toxicity of mycotoxins. Their antioxidant mechanism involves scavenging free radicals, protecting cell membranes and macromolecules from free radical damage, and maintaining cell integrity [104]. Experimental research has substantiated that the elevated polyphenol content found in edamame and various common bean varieties enhances their antioxidant characteristics. The bioactive compounds present in these beans have exhibited potent antifungal and antitoxic properties, suggesting the capacity of legumes to counteract aflatoxin contamination by leveraging the antioxidative attributes of phenolic compounds [105]. Mutagenicity resistance is reflected in reducing gene variation induced by mutagens and maintaining genetic stability [106]. This study highlights the antioxidant and anti-mutagenic potential of Piper nigrum. The ability of phytocompounds to interact with DNA might reduce the interaction of mutagens and could be one of the possible mechanisms of the anti-mutagenic activity of *P. nigrum* extract [107]. On the anti-cancer side, plant extracts can activate and induce detoxification enzymes, interfere with key oncogenic signaling pathways (such as PI3K/Akt and Raf/MEK/ERK), and suppress tumor progression [108].

Interestingly, bioactive compounds in plants have been used as additives to prevent fungal growth and mycotoxin contamination in food and feed, thereby reducing the risk of mutagenicity and carcinogenicity of mycotoxins (Table 4). For instance, the potent inhibitory effect of essential oil on a variety of molds, including *Aspergillus flavus* and *Aspergillus niger,* and their toxins has been verified. Kocic-Tanackov et al. further demonstrated that a high concentration of *Carum carvi* L. essential oil treatment can completely block the production of aflatoxin [109]. It has also been reported that calabash extract, especially the polar extract with a high ferulic acid composition, has shown potential in inhibiting toxigenic fungi and reducing mycotoxin accumulation [110]. In addition, the high efficiency of plant-derived phenols such as chlorophyll in limiting the proliferation of Fusarium verticillioides and the production of its toxin, FB1, provides a solid theoretical basis for the development of biosafety strategies to prevent the penetration of harmful mycotoxins into the food chain [111]. Phenolic substances contained in soybeans have been proven to strengthen the defense system of antifungals and inhibit the formation of aflatoxin [112]. Shah Zaman conducted experimental tests to demonstrate the potential resistance of certain plant species (*C. cyminum*). The findings revealed a statistically significant and robust negative correlation between average mycotoxins and phytochemical concentrations [113].

**Table 4 metabolites-15-00219-t004:** Overview of studies on the reduction of mycotoxin toxicity by a compound.

Plant Extract	Classification	Experimental Subject	Mechanism	References
Curcumin	Polyphenol	DucksMice	Increased jejunal tight junction protein mRNA and protein levels to protect the intestinal barrier and mitochondria from OTA-induced damageRegulation of Nrf2/p53-mediated apoptosis pathway and NF-kB/MLCK-mediated TJ pathway alleviates intestinal epithelial barrier damage induced by DON in mice	[114,115]
Resveratrol	Polyphenol	Intestinal cells	Activation of the protein kinase-dependent pathway regulates IL-6 and IL-8 secretion to promote the assembly of claudin-4 in tight junction complexes to prevent DON-induced barrier dysfunction	[116]
Dihydromyricetin	Flavone	IPEC-J2	Alleviates cell damage caused by DON through its antioxidant activity, anti-inflammatory activity, or regulation of metabolic pathways	[117]
Grape seed proanthocyanidin extract	Glycoside	Rats	GSPE can alleviate the oxidative stress induced by AFB1 and significantly improve the immune damage induced by AFB1 in mice	[118]
Red orange and lemon extract	Glycoside	Rats	RLE attenuates OTA kidney injury caused by oxidative stress	[119]
Flavonoid-rich fractions from Chromolena odorata	Flavone	Rats	Afb1-induced liver and intestinal injuries were ameliorated by changing the levels of pro-inflammatory cytokines, TNF-α, and IL-1β	[120]
Ferulic acid	Organic acid	Rats	Upregulation of tight junction proteins, downregulation of ROCK, competition for CYP450 enzyme, and activation of GST attenuate AFB1-induced duodenal barrier injury in rats	[121]
Quercetin	Polyphenol	Mice	Quercetin alleviates intestinal injury induced by DON in mice by inhibiting the TLR4/NF-κB signaling pathway and ferroptosis	[122]
Hericium mushroom polysaccharide	Polysaccharide	IPEC-J2	It can significantly protect IPEC-J2 cells from DON-induced oxidative stress, inhibit DON-induced apoptosis, and reduce the production of reactive oxygen species (ROS)	[123]
Theophylline	Alkaloid	Piglets	To improve the intestinal barrier function and reduce inflammation, immunosuppression, and oxidative stress in piglets challenged with DON by regulating NF-κB/MAPK signaling pathway	[124]
*Melaleuca alternifolia*	Essential oil	Silver catfish	Elevated levels of ROS, LOOH, and PC in plasma and liver were avoided; in addition, TTO treatment attenuated aflatoxin-related liver injury	[125]

Studies have also found that phenol-rich ginger extract (GE) can effectively reduce oxidative stress and liver injury caused by AFB1, and this protective effect may be related to the activation of the Nrf2/HO-1 signaling pathway [126]. A study demonstrated the protective effect of Tunisian radish extract (TRE) against immunosuppression induced by aflatoxin ZEN in Balb/c mice by quantifying flavonoids and isothiocyanates in TRE and measuring its antioxidant capacity [127]. Subsequently, onion, garlic extract, and eugenol were found to have significant effects on the growth and toxin synthesis of aflatoxin, especially the excellent toxin inhibition performance of onion extract [128]. In addition, carnosic acid (CA), a key polyphenol in rosemary, shows a powerful free radical scavenging effect and effectively alleviates AFB1-induced cytotoxicity and oxidative stress at the cellular level [129]. In addition, the antioxidant mechanism of total curcumin (TCMN) has been verified to significantly improve the adverse effects of AFB1 on broking chickens, restore serum biochemical indicators, and enhance the antioxidant capacity of the body, providing scientific evidence for the biological activities of the above plant extracts [130].

Scutellaria baicalensis root extract has been verified to specifically block the metabolic pathway of AFB1 in rat and human liver microsomes and reduce the generation of harmful metabolites by inhibiting CYP1A1/2 enzyme activity [131]. FWGE showed an efficient effect on alleviating the oxidative stress caused by DON and T-2 toxin in IPEC-J2 cells, reflecting its cytoprotective performance [132]. The ability of green tea extract to inhibit AFB1-induced transformation of liver cancer was confirmed across models, highlighting the value of anti-cancer components in water-soluble and ethanol extracts [133]. Agave plant extracts, especially flower parts, can significantly inhibit the production of aflatoxin and cyclopidonic acid [134]. The study of Capsantal and its mixture (Capsantal FS-30-NT) revealed its complex regulatory effects on the growth and toxin production of aflatoxin, and a low temperature was identified as the key intervention point [135]. In addition, studies have revealed that green tea extract and coumarins can regulate AFB1 metabolism in piglets and enhance detoxification, highlighting the positive utility of antioxidants in this process [136].

Currently, plant extracts have also achieved good application effects as alternatives in antimicrobial and detoxifying medications, such as common Chinese herbal medicines like licorice, bupleurum, coptis, balloon flower, isatis root, pulsatilla, and forsythia. A mixture of oregano and thyme, along with two commercial inoculants, was fermented. After 90 days of fermentation, it was indicated that the herbal extracts of oregano and thyme could be used to reduce the mycotoxin concentrations and improve the hygienic quality of corn silage [137]. The extract of Houttuynia cordata has been proven to completely replace certain antibiotics in animal diets, such as colistin sulfate or doxycycline calcium, and has been applied in large- and medium-sized enterprises for several years [138].

## 4. Limitations and Prospects

The increasing demand for plant extracts and their compounds has raised concerns about the safety, toxicity, and quality of these products. Therefore, a thorough screening of their toxicological properties is necessary. Future applications of plant extracts against mycotoxins should prioritize the development of techniques for producing safe and stable extracts. This includes comparing the similarities and differences among the active constituents of plant extracts that demonstrate effective detoxification properties and validating their precise detoxification mechanisms. Additionally, research should investigate the use of material coatings to deliver the extracts to targeted sites of action. It is also essential to determine the optimal concentrations for the application of these extracts under different conditions to minimize costs.

Research on plant extracts in the field of anti-mycotoxins has shown potential, but there are still several key gaps:(1)Most studies focus on superficial effects, such as antioxidation and anti-inflammation, but there is insufficient analysis of the molecular targets for detoxification, such as key enzyme inhibition and signaling pathway regulation, as well as the toxin metabolism pathways. There is a lack of systematic validation at the multi-omics level.(2)Active components (such as polyphenols and flavonoids) are prone to degradation under the influence of light, heat, and pH, and some extracts may cause unknown toxicities or interact with feed components. A long-term toxicological evaluation system needs to be established.(3)The extracts have low absorption rates and rapid metabolisms in animals, making it difficult to target the sites of toxin action (such as the intestines and liver). There is an urgent need to develop encapsulation technologies (such as nanocarriers) to improve delivery efficiency.(4)Existing research mostly focuses on single toxins, whereas actual contamination often involves multiple mycotoxins coexisting. The synergistic detoxification effects and mechanisms of plant extracts on complex contamination have not yet been clarified.(5)The extraction process of high-purity active components is complex and costly, and it is limited by the regional and seasonal availability of plant resources. There is a need to optimize low-cost, sustainable industrial production schemes.(6)There are no uniform quality control standards for the active components of different extracts, and there is a lack of research on the dose-effect relationship between active ingredients and detoxification effects, which affects the reliability of practical applications.

In the future, it will be necessary to combine metabolomics, targeted delivery technologies, and interdisciplinary collaboration to break through the aforementioned bottlenecks and promote the transformation of plant extracts from basic research to industrial application.

## Data Availability

No new data were created or analyzed in this study.

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
