# Peer review of "Mycotoxins in Feed: Hazards, Toxicology, and Plant Extract-Based Remedies"

_metabolites, 2025, doi:10.3390/metabo15040219_

Round 1

Reviewer 1 Report

Comments and Suggestions for Authors

Given in the report

Comments on the Quality of English Language

Given in the report

Author Response

Thank you for your suggestions on the review title. We have revised the title and incorporated a concluding remark in the abstract to emphasize the unique contributions and innovations of this review. Your comment, "Briefly provide the mechanism of mycotoxins targeting different feeds," has been addressed in the Introduction. The research gaps on plant extract-based interventions against mycotoxins, as noted in "Briefly provide the gap of the research done on plant extracts against mycotoxins," have been summarized in the Literature Review section. The additional references requested in your eighth comment have been added to lines 379-382. Other detailed revisions (lines 50-51, Table 1, lines 54-55, lines 57-58, lines 58-58, and line 64 in the revised manuscript) have also been implemented. Below are point-to-point responses to the comments.

Q1. The abstract lacks a clear hypothesis for which the review has been designed.

Response:  Thanks for your comment. We have modified the abstract accordingly. (lines 9-25 in the revised manuscript)

Q2. In the abstract and article, the authors stressed livestock and poultry production however there is no link with the article's title. I therefore suggest changing the title to"Mycotoxins in Feed: Hazards, Toxicology, and Plant Extract-based Remedies with special reference to livestock and poultry"

Response: Thanks for your comment. We have changed the title from "Mycotoxins in Feed: Hazards, Toxicology, and Plant Extract-based Remedies" to "Mycotoxins in Feed: Hazards, Toxicology, and Plant Extract-based Remedies with special reference to livestock and poultry."(lines 1-2 in the revised manuscript)

Q3. Briefly provide the mechanism of mycotoxins targeting different feeds.

Response: Thanks for your comment. The mechanism of mycotoxins targeting different feeds has been addressed in the Introduction. (lines 29-45 in the revised manuscript)

Q4. Briefly provide the gap of the research done on plant extracts against mycotoxins.

Response: Thanks for your comment. They have been summarized in the revised manuscript. (lines 438-463 in the revised manuscript)

Q5. The abstract needs conclusive remarks stating the novelty statement which make unique this review article.

Response: Thanks for your comment. We have revised the title and incorporated a concluding remark in the abstract to emphasize the unique contributions and innovations of this review. (lines 22-25 in the revised manuscript)

Q6. Biological names are always italicized, such as Aspergillus, Penicillium, Fusarium, Ergot, and Grapevine lines 32 and 33 and also in Table 1. Moreover, it is pertinent to mention that Ergot and Grapevine are not generic names.

Response: Thanks for your comment. The revisions have been made accordingly. (line50-51,table1 in the revised manuscript)

Q7. In binomial nomenclature generic names always start with capital letters such as; fusarium graminearum in Table 1 should be Fusarium graminearum.

Response: Thanks for your comment. We have revised them accordingly. (table 1 in the revised manuscript)

Q8. The authors have well explained the mechanism of different mycotoxins targeting different feed and consequently target tissues of livestock and poultry. However, there is no clear mechanism for how plant extracts and their constituents mitigate aflatoxins in feed. There is also no description of which plant extracts or their compounds have been recommended industrially. In this context the author needs to cite a study such as; DOI;10.1007/s10653-024-02045-9. This study is complementary to the current review article and will provide additional content to support your arguments.

Response: Thanks for your comment. The revisions have been made. “Shah Zaman conducted experimental tests to demonstrate the potential resistance of certain plant species (C. cyminum). The findings revealed a statistically significant and robust negative correlation between average mycotoxins and phytochemical concentrations”(lines 379-382 in the revised manuscript)

Q9. In lines # 323 and 33, Ergot is not a fungal genus instead it is a disease of fungal species of the genus Claviceps. It is therefore correct format should like, Aspergillus, Penicillium, Fusarium, Claviceps, Grapevine, and Endophytic fungi.

Response: Thanks for your comment. We have changed "such as Aspergillus, Penicillium, Fusarium, Ergot, Grapevine, and Endophytic fungi." to "such as Aspergillus, Penicillium, Fusarium, Claviceps, Grapevine, and certain species of endophytic fungi."(lines 50-51 in the revised manuscript)

Q10. Endophytic fungi are broad terms. Authors need to specify any specific endophytic fungi producing mycotoxins.

Response: Thanks for your comment. We no longer use the term endophytic fungi in the revised manuscript. (lines 50-51 in the revised manuscript)

Q11. The authors used Grapevine as a fungal genus instead it is a disease of host plant produce by fungi. Different fungi such as, Rhizopus, Euphyta, and Palsmopara are responsible fungi producing grapevine.

Response: Thanks for your comment. We have changed "such as Aspergillus, Penicillium, Fusarium, Ergot, Grapevine, and Endophytic fungi." to "such as Aspergillus, Penicillium, Fusarium, Claviceps, Grapevine, and certain species of endophytic fungi."(lines 50-51 in the revised manuscript)

Q12. There is a writing issue in line # 36 in stating pistachio along with cereals crops instead it would be better to state, (They are widely distributed in cereals such as maize, rice, and nuts such as pistachio).

Response: Thanks for your comment. We have revised the phrase“They are widely distributed in maize, rice, pistachio, cereals, and other crops” to “They are widely distributed in cereals such as maize, rice, and nuts such as pistachio.”(lines 54-55 in the revised manuscript)

Q13. In line # 39 stating thermal stability and melting point is confusing instead, authors can arrange the sentence "AFB1 is stable at most feed processing temperatures and has a boiling point of 268 °C.

Response: Thanks for your comment. We have changed the text "aflatoxin B1 (AFB1) has a melting point of 268.5°C and strong thermal stability above 100°C" to "aflatoxin B1 (AFB1) is stable at most feed processing temperatures and has a boiling point of 268 °C."(lines 57-58 in the revised manuscript)

Q14. Authors must use an alternate relevant term instead of activity of DON in line # 40 as it cannot refer to toxicity. This can be better explained by using terms such as DON remains chemically stable through processing, storage and even cooking which makes it difficult to eliminate.

Response: Thanks for your comment. We have changed the text "The activity of DON is maintained through processing, storage and even cooking, and it cannot be easily destroyed” to “DON remains chemically stable through processing, storage and even cooking which makes it difficult to eliminate.”(lines 58-59 in the revised manuscript)

Q15. In line # 46 sentence (with 60-80% of agricultural products affected) can be better structured by using "affecting 60-80% of agriculture products".

Response: Thanks for your comment. We have changed the text "with 60-80% of agricultural products affected” to “affecting 60-80% of agriculture products. (line 64 in the revised manuscript)

Reviewer 2 Report

Comments and Suggestions for Authors

The manuscript needs major revision

Author Response

Thank you very much for receiving these questions from you; some of your questions have helped me to make some additions to my review. Below are point-to-point responses to the comments.

Q1: What is the primary novel contribution of this review in comparison to prior studies on mycotoxins in feed and plant extract-based interventions?

Response: Thanks for your comment. The novel contribution lies in systematically summarizing the mechanisms by which plant extracts mitigate mycotoxin contamination and exploring their practical application potential. We also integrate recent advances, including molecular interactions between plant extracts and mycotoxins, and their impacts on animal health and productivity.

Q2: In what manner does this review fill the existing gaps in the literature about mycotoxin contamination and associated mitigation strategies?

Response: Thanks for your comment. This review fills gaps by comprehensively analyzing the detoxification efficacy of plant extracts and discussing their safety and effectiveness as mycotoxin inhibitors, thereby guiding future research directions.

Q3: Were there specific criteria for the selection of references referenced in this review?

Response: Thanks for your comment. The criteria included originality, scientific rigor, reliability, and relevance of studies.

Q4: Could you please elucidate them?

Response: Thanks for your comment. We prioritized studies from the last five years for timeliness, rigorous experimental design, large sample sizes, and robust statistical methods. Diversity and global coverage were also considered.

Q5: The manuscript addresses five principal mycotoxins—are there any emergent mycotoxins that warrant consideration as well?

Response: Thanks for your comment. Other mycotoxins are not widely existing in common feed ingredients.

Q6: Could you elucidate the molecular interactions between plant extracts and mycotoxins?

Response: Thanks for your comment. Polyphenols, flavonoids, and bioactive compounds in plant extracts bind to mycotoxins, inhibiting absorption or reducing bioactivity. Detoxification enzyme modulation is also highlighted (see end of Section 3.2 in the revised manuscript).

Q7: The review indicates antioxidant and anti-mutagenic properties of plant extracts; may additional evidence be provided?

Response: Thanks for your comment. Recent experimental studies supporting these claims have been added to Section 3.3 (first paragraph in the revised manuscript).

Q8: Do standardized techniques exist for assessing plant extract efficacy in reducing mycotoxin toxicity?

Response: Thanks for your comment.  No fully standardized methods exist, but in vitro models, animal studies, and chemical analyses are referenced.

Q9: Were geographical variations in mycotoxin contamination considered?

Response: Yes, differences due to climate, crop practices, and storage conditions are addressed (end of Section 1.2 in the revised manuscript).

Q10: What criteria were used for selecting plant extract studies, and were meta-analyses conducted?

Response: Thanks for your comment. No meta-analyses were performed; studies were selected via keyword-based searches.

Q11: Could constraints (e.g., bioavailability, stability) of plant extracts be expounded?

Response: Thanks for your comment. Constraints include chemical structure limitations, solubility, metabolic interactions, and stability during storage/processing. These are briefly noted but not elaborated.

Q12:Would comparative tables enhance clarity on mycotoxin effects across animal species?

Response: Thanks for your comment. Comparative tables have been added to improve readability.

Q13: Would figures/illustrations aid in explaining plant extract mechanisms?

Response: Thanks for your comment. However, the specific mechanisms were not fully demonstrated by the current studies.

Q14: Could real-world case studies or commercial applications be added?

Response: Case studies and commercial examples have been incorporated in the revised manuscript (end of Section 3.3 in the revised manuscript).

Q15: Are there regulatory hurdles for plant extract used in animal feed?

Response: Yes, including regulations, labeling, and safety concerns, though these are not detailed here.

Q16: What are key future research avenues?

Response: Priorities include optimizing plant extracts, clinical validation, safety assessments, and combining strategies.

Q17: Proposals for researcher-industry collaboration?

Response: Thanks for your comment. We have added the proposals to promote the researcher-industry collaboration. (lines 460-463 in the revised manuscript)

We tried our best to improve the manuscript and made some changes in the manuscript, according to the comments. These changes will not influence the framework of the paper. We appreciate for Editor's and Reviewers’ warm work earnestly and hope that the correction will meet with approval.

Once again, thank you very much for your comments and suggestions.

Best Regards,

Qian Jiang

Professor

College of Animal Science and Technology

Hunan Agricultural University, Changsha, Hunan, China

Email: jiangqian@hunau.edu.cn;

Reviewer 3 Report

Comments and Suggestions for Authors

- please check if the abstract has the number of words according to the instructions;

- the title is too general. It is not in accordance with the description in the paper. For the most part, the text makes limited reference to Effects and toxicology of 5 common mycotoxin contaminations;

- revise the head of table no. 4 (General presentation of studies on reducing the toxicity of mycotoxins by plant extracts). I suggest replacing the plant extract with a compound.

- the chapter Study on the application of plant extracts in mycotoxin poisoning is treated superficially. The names of the subchapters do not invite the reader to the place because they are too broad. for example, the importance of plant extracts...

- the names of bacteria/microorganisms are not written in italics. for example, line 324 Aspergillus Flavus. To be revised everywhere.

Author Response

We are very pleased to receive the comments from the reviewers, which have been immensely helpful. We have carefully revised the manuscript according to the comments. The revised sections were red-highlighted in the revised manuscript.

Following your recommendations, we have revised the title to "Mycotoxins in Feed: Hazards, Toxicology, and Plant Extract-based Remedies with special reference to livestock and poultry." The abstract has also been modified to align with the journal’s requirements. Regarding Table 4’s title, we have replaced "plant extracts ”with "compounds." For the subheading in Chapter 3, which was noted as lacking appeal, we have revised it to "The development potential of plant extracts." Additionally, all microbial and bacterial names requiring italicization (lines 51-52, Table 1 Column 2, line 93, line 369 in the revised manuscript) have been corrected. Below are point-to-point responses to the comments.

Q1: please check if the abstract has the number of words according to the instructions;

Response: Thanks for your comment. We have modified the abstract to increase the word count. (lines 9-25 in the revised manuscript)

Q2: the title is too general. It is not in accordance with the description in the paper. For the most part, the text makes limited reference to the Effects and toxicology of 5 common mycotoxin contaminations;

Response: Thanks for your comment. We have changed the title from "Mycotoxins in Feed: Hazards, Toxicology, and Plant Extract-based Remedies" to "Mycotoxins in Feed: Hazards, Toxicology, and Plant Extract-based Remedies with special reference to livestock and poultry."(lines 1-2 in the revised manuscript)

Q3: revise the head of table no. 4 (General presentation of studies on reducing the toxicity of mycotoxins by plant extracts). I suggest replacing the plant extract with a compound.

Response: Thanks for your comment. We have replaced "the plant extract" with "a compound".(line 383 in the revised manuscript)

Q4: the chapter Study on the application of plant extracts in mycotoxin poisoning is treated superficially. The names of the subchapters do not invite the reader to the place because they are too broad. for example, the importance of plant extracts...

Response: Thanks for your comment. We have revised “The importance of plant extracts” to "The development potential of plant extracts." (line 299 in the revised manuscript)

Q5: the names of bacteria/microorganisms are not written in italics. for example, line 324 Aspergillus Flavus. To be revised everywhere

Response: Thanks for your comment. All microbial and bacterial names requiring italicization (lines 51-52, Table 1 Column 2, line 93, line 369 in the revised manuscript) have been italicized.

We tried our best to improve the manuscript and made some changes in the manuscript, according to the comments. These changes will not influence the framework of the paper. We appreciate for Editor's and Reviewers’ warm work earnestly and hope that the correction will meet with approval.

Once again, thank you very much for your comments and suggestions.

Best Regards,

Qian Jiang

Professor

College of Animal Science and Technology

Hunan Agricultural University, Changsha, Hunan, China

Email: jiangqian@hunau.edu.cn;

Round 2

Reviewer 1 Report

Comments and Suggestions for Authors

Given in the report

Comments on the Quality of English Language

Given in the report

Reviewer 2 Report

Comments and Suggestions for Authors

Revised version of the manuscript was accepted for published 

Reviewer 3 Report

Comments and Suggestions for Authors

Thank you for modifying according to the requests and making improvements according to the suggestions.